# Translating Patent Innovation into Clinical Practice: Two Decades of Therapeutic Advancements in Dysbiosis Management

**DOI:** 10.3390/microorganisms13051064

**Published:** 2025-05-03

**Authors:** Fabiana D’Urso, Federica Paladini, Alessandro Miraglia, Alessandro D’Amuri, Marcello Chieppa, Mauro Pollini, Francesco Broccolo

**Affiliations:** Department of Experimental Medicine (DiMeS), University of Salento, 73100 Lecce, Italy; federica.paladini@unisalento.it (F.P.); alessandro.miraglia@unisalento.it (A.M.); alessandro.damuri@unisalento.it (A.D.); marcello.chieppa@unisalento.it (M.C.); mauro.pollini@unisalento.it (M.P.)

**Keywords:** dysbiosis, patent, microbiota

## Abstract

Dysbiosis, characterized by a microbial imbalance, particularly within the gut microbiota, has emerged as a significant health concern linked to various diseases. This study analyzed 8097 patent documents from The Lens database (2005–2024) to examine global innovation trends in dysbiosis management. The patent filings showed exponential growth, peaking at 1222 documents in 2022, with the United States leading in publications (4361 documents). The analysis revealed three primary innovation clusters: bacterial-based therapeutics (44.8% of patents), specific therapeutic applications (27.6%), and diagnostic methods (15.9%). The disease associations predominantly included inflammatory conditions, infections, and cancer. The patent classifications highlighted a significant focus on probiotic development and microbiota modulation. The surge in patent activity since 2014 correlates with advances in DNA sequencing technology and the growing recognition of dysbiosis’s role in human health. This analysis provides valuable insights into the evolving landscape of microbiome therapeutics and future directions for dysbiosis management.

## 1. Introduction

The human gut microbiota constitutes a complex and dynamic ecosystem of microorganisms that plays a fundamental role in health and disease. This intricate community, dominated by bacteria but also including fungi, viruses, and other microorganisms, maintains a remarkable stability in healthy adults despite daily environmental perturbations [1]. It is important to distinguish between the terms “microbiota,” which refers specifically to the microbial communities themselves, and “microbiome,” which encompasses the entire habitat, including the microorganisms, their genomes, and the surrounding environmental conditions [2]. This distinction, as outlined by Berg et al. [3], has important implications for research and therapeutic developments in the field. However, this delicate balance can be disrupted, leading to a condition known as dysbiosis, characterized by reduced microbial diversity, a loss of beneficial bacteria, and a proliferation of potentially harmful organisms [4].

The concept of dysbiosis has evolved significantly since its first documented use in 1897. The early scientific understanding of it was shaped by pioneering work in the 1960s and 1970s, particularly by researchers like Hans Haenel, who emphasized the importance of quantifying both dysbiosis and eubiosis in relation to disease [5]. This historical perspective provides a valuable context for understanding how our comprehension of dysbiosis has evolved from a simple imbalance concept to a complex condition with systemic health implications.

Recent research has revealed multiple factors that can trigger dysbiotic conditions. These include widespread antibiotic use; certain non-antibiotic medications, such as chemotherapy drugs; dietary patterns high in fat and low in fiber; sedentary behavior; circadian disruption; smoking; and early-life influences, such as the delivery mode and breastfeeding practices [6]. The clinical manifestations of dysbiosis extend far beyond gastrointestinal disturbances to include inflammatory conditions, metabolic disorders, cardiovascular disease, skin conditions, and even mental health issues [7].

Dysbiosis fundamentally represents an ecological perturbation with significant functional consequences. It manifests primarily through three key mechanisms: (1) a loss of beneficial microorganisms and their critical functions, (2) an excessive growth of potentially harmful microorganisms, and (3) a loss of overall biodiversity within the microbial community [8]. These changes can significantly impact host–microbe interactions, particularly affecting immune function, metabolic processes, and neurological signaling pathways [9]. The intestinal epithelium serves as the critical interface in this relationship, with dysbiosis often associated with increased intestinal permeability, commonly referred to as a “leaky gut,” which allows bacterial products and inflammatory mediators to enter into systemic circulation [10].

The therapeutic landscape for dysbiosis has expanded significantly, encompassing various approaches to restore the microbial balance. These include dietary interventions, probiotics, prebiotics, postbiotics, and strategies to address antibiotic resistance [11]. Symptoms may include fatigue, mental confusion, skin rashes, food intolerances, and increased susceptibility to infections. Nevertheless, several diseases and health conditions, such as cancer; inflammatory, metabolic, or autoimmune conditions; cardiovascular disease; skin conditions; and mental disorders and neurological diseases can be associated with dysbiosis [4,7,8,9,10,11,12]. The emerging treatments increasingly emphasize personalized approaches, incorporating tailored probiotics, nutritherapy, functional foods, and advanced microbiota transplantation techniques. The development of these interventions reflects a growing understanding of the individual variations in microbiome composition and response to treatment.

The therapeutic strategies that have been discussed for dysbiosis include the restoration of the microbial balance through dietary action and the use of probiotics, prebiotics, and postbiotics [4,7,12,13]. Probiotics, defined as “live microorganisms that, when administered in adequate amounts, confer a health benefit on the host” [14], have become increasingly sophisticated, evolving from simple single-strain formulations to complex multi-strain preparations designed to address specific dysbiotic conditions [15]. Prebiotics, non-digestible food ingredients that selectively stimulate the growth or activity of beneficial bacteria, have similarly advanced, with the development of increasingly targeted compounds that promote the growth of specific beneficial bacterial populations [16]. Postbiotics, which include bacterial metabolites, cell components, and other bioactive compounds derived from microorganisms, represent an emerging therapeutic avenue with significant potential for addressing dysbiosis-related conditions [17].

The control of antibiotic-resistant bacteria has also been studied [18]. These strategies underscore the intersection of scientific research and technology for addressing microbiome-related disorders by developing innovative solutions and inventions. Recent research has also explored the potential of bacteriophage therapy as a precision tool for modulating microbiota composition without the collateral damage often associated with antibiotic treatment [19]. This approach offers particular promise for targeting specific pathogenic bacteria while preserving beneficial microbial populations, representing a significant advance in dysbiosis management strategies.

The surge in microbiome research has been accompanied by significant technological innovation, particularly evident in the patent landscape. Patents, as legal documents granting exclusivity rights for novel methods and formulations, can provide valuable insight into the state-of-the-art and development trends in dysbiosis management. The intellectual property system, through patent documentation, provides crucial protection for these scientific advances. These legal instruments serve as both shields for original discoveries and detailed records of technological progress. When researchers examine patent collections, including both those approved and under review, they gain valuable insights into scientific advancement and emerging trends. Patent analysis reveals not only technological evolution but also illuminates the complex network of scientific teams, research institutions, and commercial enterprises driving innovation forward. The scope of protected technologies becomes clear through both the detailed technical reviews and standardized classification systems employed by patent authorities worldwide.

The strategic significance of addressing dysbiosis has become increasingly apparent through recent evaluations of patent trends in functional nutrition. A comprehensive review by Matin et al. in 2024 [20] highlighted how microbiome-modulating compounds, particularly probiotics and prebiotics, have emerged as central themes in this field. The patent landscape reflects diverse technological approaches, from laboratory-engineered beneficial bacteria targeting specific harmful microorganisms (WO2020/139852A1) to specialized dietary formulations promoting intestinal health (WO2018/023003A1). The field has also seen advances in diagnostic capabilities, with new tools for detecting dysbiosis markers (US2018/0267037A1 and CN110097928B) and sophisticated platforms for microbiome analysis that support personalized treatment strategies (US2023/245733A1). These protected innovations demonstrate a growing commercial commitment to transforming microbiome research into practical medical solutions.

To better understand this dynamic field, we conducted an extensive review of dysbiosis-related patents using The Lens database (https://www.lens.org, accessed on 26-04-2025), a comprehensive repository of global patent information. Our analysis offers researchers a panoramic view of the technological developments in dysbiosis management, highlighting the key trends and potential future directions in this rapidly evolving field.

## 2. Materials and Methods

Our study employed The Lens patent database, containing over 159 million patent documents, to gather comprehensive data on dysbiosis-related innovations. We conducted a search using the term “dysbiosis” in all patent fields, covering January 2005 through October 2024. The database was accessed on 15 February 2024 to ensure reproducibility of our findings. This query yielded 8097 relevant patent documents, including both applications and granted patents.

We analyzed temporal trends in patent publications, geographical distribution patterns, and ownership structures. Temporal analysis examined publication trends over time, tracking both overall volume and the proportion of granted versus pending patents. This analysis included assessment of compound annual growth rates and identification of key inflection points in innovation activity.

Geographic distribution analyzed the jurisdictional distribution of patent filings, with particular attention to leading countries and regions. This analysis considered both primary filings and subsequent international extensions.

Disease associations were mapped through title analysis of documents grouped into simple families to avoid redundancy. This analysis focused on both direct therapeutic targets and broader health implications.

Citation network analysis examined both patent-to-patent and patent-to-literature citations to identify influential innovations and underlying scientific foundations. Citation patterns were analyzed to understand knowledge flow and technological evolution. Citation networks were examined to identify influential patents and scientific publications.

Additionally, we conducted detailed examination of International Patent Classification (IPC) codes to understand technological focus areas. IPC code analysis included detailed examination of International Patent Classification codes to understand technological focus areas and their evolution over time. This analysis included both primary and secondary classification codes to capture the full scope of innovations.

Data processing utilized Python version 3.9.7 for textual analysis, with particular attention to disease term frequency and technological classifications. Specific packages employed included pandas (1.3.4) for data manipulation, matplotlib (3.4.3) and seaborn (0.11.2) for visualization, scikit-learn (1.0) for statistical analysis, and NetworkX (2.6.3) for network analysis. Natural language processing (NLP) was performed using NLTK (3.6.5) for text preprocessing and spaCy (3.2.0) for entity recognition. Statistical analysis of temporal trends employed standard bibliometric methods, including time series analysis using the statsmodels package (0.13.0). Patent family analysis helped avoid multiple counting of essentially identical innovations filed in different jurisdictions.

For statistical analysis, we employed descriptive statistics to summarize patent activity trends, including measures of central tendency and dispersion. Compound annual growth rate (CAGR) was calculated using the formula CAGR = (EV/BV) ^ ^(1/n) − 1^, where EV represents end value, BV represents beginning value, and n represents the number of years. Correlation analysis was performed using Pearson’s correlation coefficient to assess relationships between patent activity and external factors, such as research funding and technological advances. Statistical significance was set at *p* < 0.05 for all analyses.

## 3. Results

### 3.1. Temporal Analysis and Geographic Distribution

The patent filings related to dysbiosis showed remarkable growth over the studied period. From just 6 documents in 2005, the number of publications grew steadily until 2014, when the annual output first exceeded 100 documents. This growth accelerated significantly thereafter, reaching 1222 publications in 2022 (Figure 1A), representing a compound annual growth rate of 36.8% over the full period.

Currently, 37% of the published documents maintain an active status (Figure 1B), while 50% remain under examination. This high proportion of pending and active patents reflects the relative youth of innovation in this field, suggesting a continued growth potential. The remaining 13% of the patents have expired or been withdrawn, primarily due to technical or procedural reasons rather than scientific merit.

The geographical analysis revealed the United States as the dominant jurisdiction, accounting for 4361 patent documents (53.9% of the total publications) (Figure 1C). This leadership extends to corporate involvement, with seven of the top ten patent-holding companies headquartered in the United States. The concentration of innovation in the U.S. correlates with substantial research investments, particularly from specialized biotech startups receiving significant venture capital funding. Europe follows as the second-most-active region (1962 documents, 24.2%), with China (832 documents, 10.3%) showing the fastest growth rate in recent years. This geographic distribution reflects both the concentration of research infrastructure and the strategic importance different regions place on microbiome research and its commercial applications.

### 3.2. Corporate Leadership and Innovation Focus

Psomagen Inc. (Rockville, MD, USA) leads patent holdings with 171 documents (Figure 1D), focusing primarily on genomics solutions and clinical sequencing. Their portfolio expanded significantly following the 2020 acquisition of Ubiome Inc.’s intellectual property, which held the second-largest patent collection (155 documents). The third-largest holder, CJ Bioscience Inc. (Seoul, South Korea), with 146 documents, specializes in AI-based microbiome drug development.

Other significant patent holders include Seres Therapeutics (128 documents), focusing on microbiome therapeutics for infectious and inflammatory diseases; Vedanta Biosciences (112 documents), specializing in defined bacterial consortia; and Finch Therapeutics (97 documents), developing microbiota-based drugs for treating recurrent *C. difficile* infection. This corporate landscape demonstrates both the commercial potential of dysbiosis-related innovations and the diverse approaches being pursued by industry leaders.

### 3.3. Disease Association Analysis

The analysis of the patent titles revealed inflammatory diseases as the most frequently cited condition (197 citations), followed by infectious diseases (138 citations), cancers (126 citations), gastrointestinal disorders (112 citations), and skin conditions (98 citations) (Figure 2A). This distribution aligns with the current understanding of dysbiosis’s systemic effects, as documented by Thomas et al. (2023) and Dréno et al. (2020) [8,9].

Neurological conditions (76 citations) and metabolic disorders (63 citations) also emerged as significant focus areas, reflecting the growing recognition of the gut–brain axis and the metabolic implications of dysbiosis [21]. The relatively high proportion of patents addressing multiple conditions (47 citations) suggests an increasing appreciation of dysbiosis’s role in complex, multi-system disorders and the potential for broadly applicable therapeutic approaches.

The most cited patent document (US2014/0147425A1, 346 citations) describes therapeutic compositions containing purified bacterial populations for preventing and treating dysbiosis symptoms (Figure 2B). The other highly cited patents focus on bacterial combinations for immune regulation (WO2015/095241A2, 287 citations) and genetic material manipulation for diagnostics (US8603749B2, 245 citations).

### 3.4. Technological Classification Analysis

The IPC code analysis revealed three primary innovation clusters. Bacterial-based therapeutics emerged as the dominant focus, represented by code A61K35/74 and the related subcodes in 3632 documents (44.8% of the total). This cluster encompasses the development of probiotics (A61K35/741), spore-forming bacteria (A61K35/742), and various beneficial bacterial strains (A61K35/744-747).

Specific therapeutic applications formed the second-largest cluster with 2231 documents (27.6%), addressing digestive system disorders (A61P1/00), antibacterial applications (A61P31/04), and anti-inflammatory treatments (A61P29/00).

Diagnostic and analytical methods were the subject of 1291 documents (15.9%), focusing on nucleic acid measurements (C12Q1/68) and bacterial identification techniques (C12Q1/689).

The remaining patents (11.7%) covered various supplementary technologies, including culture media development (C12N1/20) and food modification techniques (A23L33/00).

This technological distribution reflects the field’s primary focus on developing direct interventions for dysbiosis, with a secondary emphasis on diagnostic tools and enabling technologies. The predominance of bacterial-based approaches suggests continued confidence in the modulation of the microbiota itself as the most promising therapeutic strategy.

### 3.5. Scientific Impact Analysis

The most cited scientific reference in the patent documents (219 citations) focused on the molecular–phylogenetic characterization of the microbial communities in inflammatory bowel diseases, reflecting the importance of precise microbial identification. This aligns with recent advances in DNA sequencing technology that have revolutionized microbiome analysis [22,23].

Other highly cited works emphasized the role of dysbiosis in specific conditions, particularly inflammatory bowel disease [23] and various systemic disorders [24]. This citation pattern demonstrates the broadening understanding of dysbiosis’s impact on human health.

### 3.6. Leading Products from Key Patents

To provide a clinical context for the patent analysis, we examined the most influential patents based on their citation frequency and identified the principal commercial products developed from these innovations. This analysis reveals the translation of patent-protected technologies into clinical applications.

#### 3.6.1. SER-109 (Seres Therapeutics)

Developed under US2014/0147425A1 (346 citations), SER-109 represents a pioneering microbiome therapeutic comprising purified bacterial spores from healthy donor stool. This product completed Phase III clinical trials, demonstrating an 88% efficacy for preventing recurrent *C. difficile* infection, significantly outperforming the placebo (60%) [25]. SER-109 received FDA approval in April 2023 as VOWST™, becoming the first FDA-approved live biotherapeutic product. The preparation process involves rigorous donor screening, spore purification, and safety testing to eliminate pathogens while preserving the therapeutic bacterial populations. Clinical studies have demonstrated that SER-109 effectively restores microbiome diversity and functionality, establishing colonization resistance against *C. difficile* [26].

#### 3.6.2. VE303 (Vedanta Biosciences)

Based on WO2015/095241A2 (287 citations), VE303 consists of a defined bacterial consortium designed to restore colonization resistance against *C. difficile*. Unlike stool-derived products, VE303 features a precisely defined mix of eight bacterial strains grown from pure isolates under controlled conditions [27]. The phase II clinical trials demonstrated an 86.7% sustained clinical cure at 8 weeks post-treatment with high-dose VE303, compared to 42.9% with the placebo. The product’s defined composition offers significant manufacturing and regulatory advantages, contributing to its consistent therapeutic effects and improved safety profile. VE303 represents a significant advance in precision microbiome therapeutics, moving away from undefined fecal transplant approaches toward standardized, reproducible treatments [28].

#### 3.6.3. IBDiagnostics Platform (GeneCentric Therapeutics)

Developed from US8603749B2 (245 citations), this diagnostic platform employs genetic markers to assess the dysbiosis patterns characteristic of inflammatory bowel disease (IBD). The system analyzes key microbial signatures to differentiate between Crohn’s disease and ulcerative colitis with 87% accuracy, exceeding that of traditional serological markers [29]. Clinical validation studies have demonstrated a significant improvement in diagnostic clarity when combined with a standard clinical assessment, reducing the diagnostic uncertainty by 28% [30]. This platform exemplifies how patent-protected diagnostic technology can improve clinical decision-making and potentially guide personalized therapeutic approaches for IBD patients.

#### 3.6.4. CP101 (Finch Therapeutics)

Based on WO2017/079450A1 (197 citations), CP101 represents an orally administered, full-spectrum microbiome therapeutic for recurrent *C. difficile* infection. The product uses an encapsulated, freeze-dried preparation of donor-derived microbiota designed to restore gut ecosystem function [31]. The Phase II clinical trials demonstrated a 74.5% efficacy at 8 weeks, with sustained benefits through 24 weeks post-treatment. CP101’s development reflects a hybrid approach using undefined fecal microbiota transplantation and defined consortia, preserving ecological complexity while improving the delivery and patient acceptability through an oral capsule formulation [32].

#### 3.6.5. Evelo Biosciences EDP1815

Protected under WO2018/089789A1 (172 citations), EDP1815 utilizes a single strain of *Prevotella histicola* to modulate inflammatory pathways. This oral monoclonal microbial product acts locally in the small intestine to produce systemic anti-inflammatory effects without systemic exposure to the organism [33]. The Phase II clinical trials for psoriasis demonstrated a significant improvement in the PASI scores (Psoriasis Area and Severity Index) compared to the placebo, with an excellent safety profile comparable to that of the placebo [34]. The product exemplifies how single-strain approaches can elicit specific immunomodulatory effects, addressing the inflammatory manifestations of dysbiosis without attempting to reconstruct the entire microbiome.

#### 3.6.6. Diagnostic Platforms Based on Microbial Metabolites

Several products that have been developed under US2018/0267037A1 (158 citations) focus on metabolite-based dysbiosis assessment rather than direct microbial identification. These platforms analyze short-chain fatty acids, bile acids, and other microbial metabolites as functional indicators of microbiome health [35]. The clinical validation studies have demonstrated an 83% concordance with traditional culture-based assessments, while providing additional insights into the functional implications of dysbiosis [36]. This approach represents an important complement to genomic-based diagnostics, focusing on functional outcomes rather than merely the taxonomic composition.

This product analysis demonstrates the diverse clinical applications emerging from patent-protected innovations in dysbiosis management. From defined bacterial consortia to metabolite-based diagnostics, these products represent significant advances in translating microbiome science into practical clinical tools.

## 4. Discussion

The patent landscape analysis reveals several significant implications for the future of dysbiosis management. The exponential growth in patent filings since 2014 has coincided with major technological advances, particularly in DNA sequencing and microbiome analysis techniques [22]. This growth reflects not only an improved understanding of dysbiosis’s role in various diseases, but also the increasing commercial confidence in microbiome-based therapeutics.

The American dominance in the patent landscape can be attributed to several factors. First, substantial research investments, particularly from specialized biotech startups, have accelerated innovation. Second, a well-developed intellectual property system and regulatory framework have facilitated patent protection. Third, strong academic–industry collaboration has helped translate basic research into practical applications.

The technological focus on bacterial-based interventions, evidenced by the prevalence of IPC code A61K35/74, suggests an evolution toward more sophisticated therapeutic approaches. The early patents focused primarily on traditional probiotics, while recent innovations have increasingly involved engineered bacterial strains and targeted delivery systems. This progression indicates a growing appreciation for the complexity of microbiome modulation.

The distribution of disease associations in the patent documents aligns with the emerging clinical understanding of dysbiosis’s systemic effects. The prominence of inflammatory conditions reflects the growing evidence linking microbial imbalance to immune dysfunction [23]. Similarly, the focus on infectious diseases reflects an increasing awareness of how dysbiosis affects pathogen susceptibility [24].

The emerging areas of innovation, particularly in diagnostic methods and food modification, suggest a trend toward more personalized approaches to dysbiosis management. The integration of artificial intelligence and advanced analytical techniques indicates the potential for more precise intervention strategies. Additionally, the growing attention to nutritional approaches reflects an understanding of the diet’s role in maintaining microbiome health.

This close examination of the clinical products that have been derived from the leading patents reveals important translational progress in the field. The evolution from the use of undefined fecal microbiota transplantation to precisely defined bacterial consortia represents a significant maturation of the therapeutic approach. Products like SER-109 and VE303 demonstrate how rigorous clinical development can transform patent-protected innovations into approved therapeutics with demonstrated efficacy [25,27]. Similarly, the development of sophisticated diagnostic platforms based on genetic markers and microbial metabolites illustrates how patent-protected technologies can enhance clinical decision-making [29,35].

This translation from patent to product has accelerated in recent years, with regulatory agencies developing specialized frameworks for microbiome-based therapeutics. The FDA’s approval of VOWST™ (SER-109) in 2023 established an important regulatory precedent, potentially smoothing the path for subsequent microbiome therapeutics [26]. This regulatory evolution parallels the technological maturation evident in the patent landscape, with both realms showing increasing sophistication and specialization.

The patent landscape also reveals several challenges and opportunities. While bacterial-based therapeutics dominate the current innovations, there is a growing recognition of the need for more holistic approaches that consider the broader ecological context of the microbiome. Additionally, the emergence of patents focusing on specific patient subgroups suggests a movement toward more targeted therapeutic strategies.

The current patent trends indicate a growing interest in precision approaches to dysbiosis management. Recent innovations have increasingly focused on identifying the specific microbial signatures associated with particular disease states, enabling more targeted interventions [37]. Additionally, patents addressing the role of microbial metabolites in disease pathogenesis suggest a growing appreciation for the functional implications of dysbiosis beyond simple taxonomic changes [38].

Another significant trend is the increasing focus on combination therapies that address multiple aspects of dysbiosis simultaneously. Recent patents have explored the synergistic effects of probiotics and prebiotics, as well as combinations of microbiome-modulating agents with traditional pharmaceuticals [39]. This integrated approach reflects a growing understanding of dysbiosis as acomplex ecological disturbance requiring multifaceted intervention strategies.

The emergence of microbial ecosystem engineering as a patent focus area represents, perhaps, the most ambitious direction in the field. Recent innovations have explored methods for precision editing of the microbiome, including targeted bacteriophage therapy, engineered bacterial consortia with specific ecological functions, and tools for the selective depletion of harmful bacteria [40]. These approaches have moved beyond simply adding beneficial bacteria to actively reshaping the microbial ecosystem, representing a significant evolution in therapeutic philosophy.

This comprehensive analysis of the dysbiosis patent landscape reveals a dynamic field characterized by rapid technological evolution and growing commercial interest [41,42,43,44,45,46,47,48,49,50]. The predominance of bacterial-based therapeutic approaches, coupled with an increasing focus on diagnostic precision and personalized interventions, suggests a maturing understanding of dysbiosis management. Several key trends emerge as particularly significant for future development:The integration of advanced technologies, particularly artificial intelligence and precise microbial identification methods, is likely to enable more targeted therapeutic approaches.The growing emphasis on personalized interventions reflects the recognition of individual variations in microbiome composition and response to treatment.The increasing focus on preventive strategies, particularly through dietary modifications and early-life interventions, suggests an evolution toward more proactive management approaches.

The knowledge gained through this analysis can help accelerate the development of effective solutions, inform regulatory and market strategies, and facilitate the translation of microbiome science into practical health interventions. As our understanding of dysbiosis continues to evolve, the patent landscape will likely expand further, potentially revealing new therapeutic approaches and technological innovations in microbiome health management.

## 5. Conclusions

This patent landscape analysis illuminates several promising directions for future research and development in dysbiosis management. Advanced diagnostic technologies represent a fertile area for innovation, with significant potential for rapid, point-of-care testing methods and the integration of multi-omics approaches combining metabolomics, proteomics, and metagenomics. Machine learning algorithms for predictive dysbiosis modeling show particular promise, as does the development of comprehensive biomarker panels for the early detection of dysbiosis-related conditions. Innovation in sampling techniques continues to advance, promising the improved accuracy of microbiome analyses.

Therapeutic innovations represent another critical frontier, with the engineering of precision probiotics targeting specific dysbiotic conditions emerging as a key focus area. The development of microbiome-based drug delivery systems and the investigation of bacteriophage therapy for selective microbiome modulation demonstrates the field’s increasing sophistication. The research into synthetic biology approaches for microbiome engineering is advancing rapidly, complemented by novel prebiotic compounds with enhanced specificity and the integration of immunomodulatory strategies with microbiome therapeutics.

Prevention and monitoring strategies are evolving in parallel, focusing on preventive interventions for high-risk populations and the creation of real-time monitoring systems for microbiome stability. Healthcare providers are increasingly implementing personalized treatment protocols based on individual microbiome profiles, supported by standardized dysbiosis assessment tools and the integration of microbiome analysis into routine health screenings.

Specialized treatment centers focusing on dysbiosis management are emerging, while clinical decision support systems incorporating microbiome data are becoming more sophisticated. The preventive medicine applications are expanding rapidly, with early-life interventions to prevent dysbiosis gaining particular attention alongside workplace and community-based prevention programs.

The commercial applications are diversifying, with consumer-focused microbiome testing services and personalized nutrition programs based on microbiome profiles leading the way. Innovations in functional foods and supplements continue to accelerate, while microbiome-friendly personal care products represent an emerging market opportunity.

Technical challenges persist, particularly regarding the standardization of microbiome analysis methods and the maintenance of the long-term therapeutic effects, while regulatory considerations are evolving in parallel with technical innovations. Market development faces its own challenges, with the education of healthcare providers and consumers representing a crucial focus area.

The future success of dysbiosis management will depend on continued innovation, collaboration among stakeholders, and careful attention to both technical and practical implementation challenges, ultimately working toward the comprehensive integration of dysbiosis management into standard medical practice.

## Figures and Tables

**Figure 1 microorganisms-13-01064-f001:**
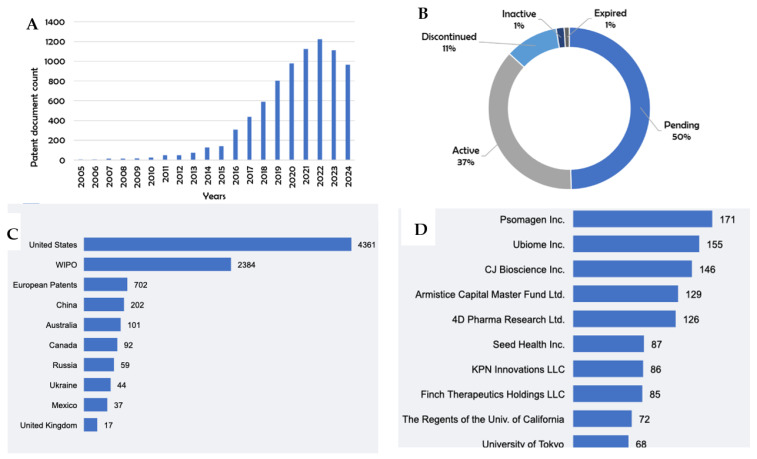
(**A**) Evolution of dysbiosis patent publications since 2005 (year 2024 data are limited to those published up until 28 October); (**B**) current legal status of dysbiosis patent documents; (**C**) top 10 dysbiosis patent documents jurisdictions; and (**D**) top 10 dysbiosis patent documents owners.

**Figure 2 microorganisms-13-01064-f002:**
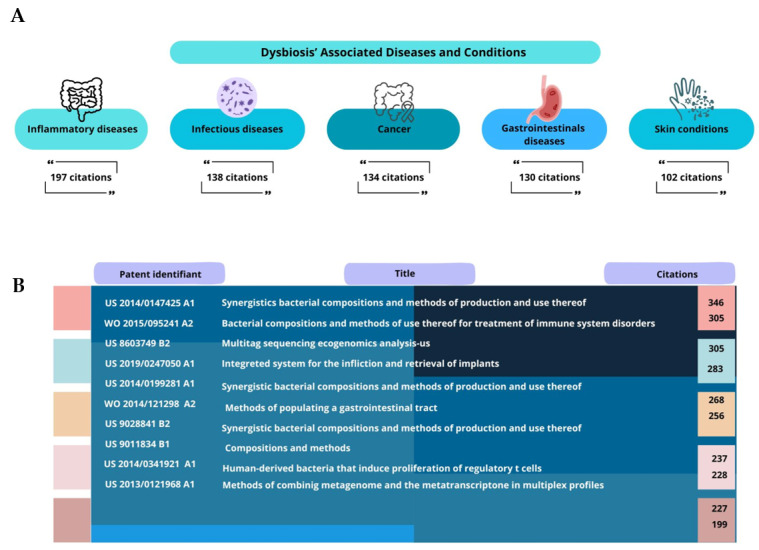
(**A**) The most dysbiosis-associated diseases and; (**B**) the top 10 most cited dysbiosis patent documents.

## Data Availability

The data are available on request.

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
