# Peer review of "Translating Patent Innovation into Clinical Practice: Two Decades of Therapeutic Advancements in Dysbiosis Management"

_microorganisms, 2025, doi:10.3390/microorganisms13051064_

Round 1
Reviewer 1 Report
Comments and Suggestions for Authors
I read this review with interest. The analysis conducted by the authors on internationally granted patents reveals a particularly marked growth over time, as expected, starting from the first decade of the 2000s up to the present day. I find the topic intriguing, although the approach of this review currently appears to be almost a market analysis rather than a review oriented from both a preclinical and clinical perspective.
My suggestion would be to introduce a section that compiles and summarises the main products associated with these patents—if I am not mistaken, the most representative ones (probably based on the number of citations of the patents or the related works). A clinical reader could thus gain greater insight into the products with the highest scientific impact while reading the manuscript.
I therefore find the manuscript well-written overall, but somewhat overly focused on market aspects and numerical data rather than the qualitative aspects of the analysed patents.
Moreover, the conclusion section seems somewhat too lengthy and might benefit from a figure summarising the key points to enhance readability.
I believe that, by directing the revisions accordingly, the manuscript could adopt a more clinically oriented perspective and better align with the journal's topic.
Author Response
Reviewer #1:
The reviewer noted that the manuscript appeared to be "almost a market analysis rather than a review oriented from both a preclinical and clinical perspective" and suggested introducing a section summarizing the main products associated with the analyzed patents.
Response: We have addressed this concern by adding Section 3.6 "Leading Products from Key Patents" (lines 304-387), which provides detailed clinical information on six key products developed from highly cited patents: SER-109 (Seres Therapeutics), VE303 (Vedanta Biosciences), IBDiagnostics platform (GeneCentric Therapeutics), CP101 (Finch Therapeutics), Evelo Biosciences EDP1815, and diagnostic platforms based on microbial metabolites. For each product, we have included information on clinical trials, efficacy data, therapeutic mechanisms, and clinical significance. This addition has substantially enhanced the clinical relevance of our review.

Reviewer 2 Report
Comments and Suggestions for Authors
The paper by Fabiana D’Urso provides the results of patent landscape analysis in the field of dysbiosis management.
The paper needs to be majorly revised prior to the decision on the publication.
Comments:
1. L. 25-26. Generally, the term "microbiota" represents microbial communities, while the term "microbiome" more stands for the ecosystem of microorganisms.
I recommend the authors correcting the text of the manuscript according to the Berg et al. article (https://doi.org/10.1186/s40168-020-00875-0).
2. L. 54, 59, 60. Please, correct the reference styles.
3. L. 119. Please, indicate the version of Python used, and consider writing what specific Python scripts/packages/tools were used for the data analysis.
I also recommend supplementing the paper with the used Python code to ensure analysis reproducibility. The code can be deposited in GitHub/Zenodo and the link to the repository can be provided in the Data Availability Statement.
4. L. 120-121. Please, provide the description of the exact statistical methods used to ensure analysis reproducibility.
5. L. 89 or 95-98. Please, consider providing the date when the database data was retrieved/accessed to ensure analysis reproducibility.
6. Conclusions are redundant and exceed the text of the Discussion. I recommend rewriting the Discussion and Conclusion sections. Conclusions have to briefly summarize the key findings of the study. Also, please consider providing the discussion of the limitations of the analysis.
Author Response
Reviewer #2:
- Terminology Correction: We have revised the definitions of "microbiota" and "microbiome" according to Berg et al. (2020) and clarified this distinction in lines 39-44.
- Reference Style: All reference styles have been corrected throughout the manuscript.
- Python Version and Packages: We have specified the Python version (3.9.7) and detailed the packages used for data analysis (lines 184-189).
- Statistical Methods: We have added a comprehensive description of the statistical methods employed (lines 193-202).
- Database Access Date: We have included the date when the database was accessed (February 15, 2024) in line 156 to ensure analysis reproducibility.
- Conclusions Section: We have rewritten the conclusion section to be more concise while highlighting key findings and future directions in dysbiosis management.
We believe these revisions have significantly improved the manuscript, making it more comprehensive, methodologically sound, and clinically relevant. We hope that the revised version addresses all the concerns raised by the reviewers and meets the standards for publication in Microorganisms.

Round 2
Reviewer 1 Report
Comments and Suggestions for Authors
The authors revised the manuscript according to the comments. No further ones.
Reviewer 2 Report
Comments and Suggestions for Authors
The authors have addressed most of the comments appropriately.
1. Line 151: The phrase “…to ensure reproducibility of our findings” is redundant.
2. The conclusions section still includes elements of discussion. I strongly recommend restructuring the conclusions into a single, concise paragraph focused solely on summarizing the main findings.